# Fine-grained Few-shot Recognition by Deep Object Parsing

## Abstract

We propose a new method for fine-grained few-shot recognition via deep object parsing. In our framework, an object is made up of $K$ distinct parts and for each part, we learn a dictionary of templates, which is shared across all instances and categories. An object is parsed by estimating the locations of these $K$ parts and a set of active templates that can reconstruct the part features. We recognize test instances by comparing its active templates and the relative geometry of its part locations against those of the presented few-shot instances. Our method is end-to-end trainable to learn part templates on-top of a convolutional backbone. To combat visual distortions such as orientation, pose and size, we learn templates at multiple scales, and at test-time parse and match instances across these scales. We show that our method is competitive with the state-of-the-art, and by virtue of parsing enjoys interpretability as well.

## 1 Introduction

Deep neural networks (DNN) can be trained to solve visual recognition tasks with large annotated datasets. In contrast, training DNNs for few-shot recognition (Snell et al., 2017; Vinyals et al., 2016), and its fine-grained variant (Sun et al., 2020), where only a few examples are provided for each class by way of supervision at test-time, is challenging. Fundamentally, the issue is that few-shots of data is often inadequate to learn an object model among all of its myriad of variations, which do not impact an object's category. For our solution, we propose to draw upon two key observations from the literature.

(A) There are specific locations bearing distinctive patterns/signatures in the feature space of a convolution neural network (CNN), which correspond to salient visual characteristics of an image instance (Zhou et al., 2014; Bau et al., 2017).

(B) Attention on only a few specific locations in the feature space, leads to good recognition accuracy (Zhu et al., 2020; Lifchitz et al., 2021; Tang et al., 2020).

**How can we leverage these observations?**
*Duplication of Traits.* In fine-grained classification tasks, we posit that the visual characteristics found in one instance of an object are widely duplicated among other instances, and even among those belonging to other classes. It follows from our proposition that it is the particular collection of visual characteristics arranged in a specific geometric pattern that uniquely determines an object belonging to a particular class.

*Parsing.* These assumptions, along with (A) and (B), imply that these shared visual traits can be found in the feature maps of CNNs and only a few locations on the feature map suffice for object recognition. We call these finitely many latent locations on the feature maps which correspond to salient traits, *parts*. These parts manifest as patterns, where each pattern belongs to a finite (but potentially large) dictionary of templates. This dictionary embodies both the shared vocabulary and the diversity of patterns found across object instances. Our goal is to learn the dictionary of templates for different parts using training data, and at test-time, we seek to *parse*[1] new instances by identifying part locations and the sub-collection of templates that are expressed for the few-shot task. While CNN features distill essential information from images, parsing helps further suppress noisy

---

[1] we view our dictionary as a collection of words, parts as phrases that are a collection of words from the dictionary, and the geometric relationship between different parts as relationship between phrases.

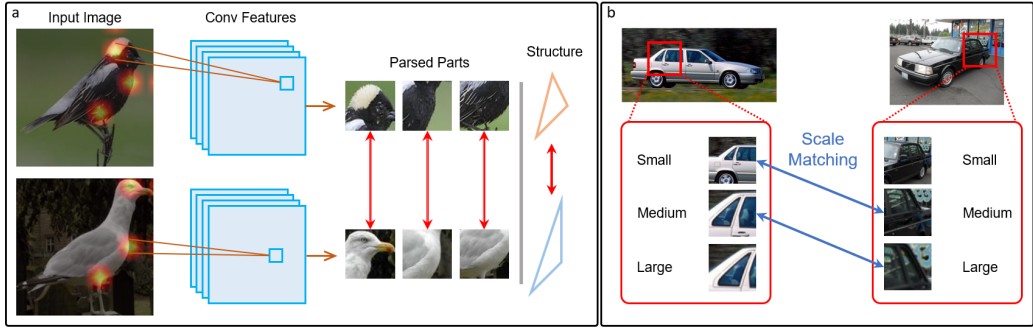

Figure 1: **Motivation:** a) In fine-grained few-shot learning, the most discriminating information is embedded in the salient parts (e.g. head and breast of a bird) and the geometry of the parts (relative part locations). Our method parses the object into a structured combination of a finite set of dictionaries, such that both finer details and the shape of the object are captured and used in recognition. b) In few shot learning, the same part may be distorted or absent in the support samples due to the perspective and pose changes. We propose to extract features and compare across multiple scales for each part to overcome this.

information, in situations of high-intra class variance such as in few-shot learning. For classification, few-shot instances are parsed and then compared against the parsed query. The best matching class is then predicted as the output.

As an example see Fig 1 (a), where the recognized part locations using the learned dictionary correspond to the head, breast and the knee of the birds in their images with corresponding locations in the convolutional feature maps. In matching the images, both the constituent templates and the geometric structure of the parts are utilized. Inferring part locations based on part-specific dictionaries is a low complexity task, and is analogous to the problem of detection of signals in noise in radar applications (Van Trees, 2004), a problem solved by matching the received signal against a known dictionary of transmitted signals.

*Challenges.* Nevertheless, our situation is somewhat more challenging. Unlike the radar situation, we do not a-priori have a dictionary, and to learn one, we are only provided class-level annotations by way of supervision. In addition, we require that these learnt dictionaries are compact (because we must be able to reliably parse any input), and yet sufficiently expressive to account for diversity of visual traits found in different objects and classes.

*Multi-Scale Dictionaries.* Variations in position and orientation relative to the camera lead to different appearances of the same object by perspective projections, which means there is variation in the sizes of visual characteristics of parts. To overcome this, we train dictionaries at multiple scales, which leads us to a parsing scheme that parses input instances at multiple scales (see Fig. 1 (b)).

*Goodness of fit.* Besides part sizes, few-shot instances even within the same class may exhibit significant variations in poses, which can in-turn induce variations in parsed outputs. To mitigate their effects we propose a novel instance-dependent re-weighting method, for comparison, based on goodness-of-fit to the dictionary.

*Contributions.* (i) We propose a deep object parsing (DOP) method that parses objects into its constituent parts, and each part as a collection of activated templates from a dictionary, while using the representational power of deep CNNs. Via suitable objectives, we derive a simple end-to-end trainable formulation for this method. (ii) We evaluate DOP on the challenging task of fine-grained few shot recognition, where DOP outperforms prior art on multiple benchmarks. Notably, it is better by about 2.5% on Stanford-Car and 10% on the Aircraft dataset. (iii) We provide an analysis of how different components of our method help final performance. We also visualize the part locations recognized by our method, lending interpretability to its decisions.

## 2 RELATED WORK

**Few-Shot Classification (FSC).** Modern FSC methods can be classified into three categories: metric-learning based, optimization-based, or data-augmentation methods. Methods in the first cat-

egory focus on learning effective metrics to match query examples to support. Prototypical Network (Snell et al., 2017) utilizes euclidean distance on feature space for this purpose. Subsequent approaches built on this by improving the image embedding space (Ye et al., 2020; Das et al., 2021; Afrasiyabi et al., 2021; Zhou et al., 2021; Rizve et al., 2021) or focusing on the metric (Sung et al., 2018; Wang et al., 2019; Bateni et al., 2020; Simon et al., 2020; Li et al., 2020a; Wertheimer et al., 2021; Fei et al., 2021; Zhang et al., 2020; 2021). Some recent methods have also found use of graph based methods, especially in transductive few shot classification (Chen et al., 2021; Yang et al., 2020). Optimization based methods train for fast adaptation using a few parameter updates with the support examples (Finn et al., 2017; Baik et al., 2021; Li et al., 2017; Lee et al., 2019; Rajeswaran et al., 2019; Liu et al., 2018). Data-augmentation methods learn a generative model to synthesize additional training data for the novel classes to alleviate the issue of insufficient data (Li et al., 2020b; Schwartz et al., 2018; Wang et al., 2018; Xu et al., 2021).

**Fine-grained FSC.** In fine-grained few-shot classification, different classes differ only in finer visual details. An example of this is to tease apart different species of birds in images. The approaches mentioned above have been applied in this context as well (Li et al., 2019; Sun et al., 2020; Li et al., 2020c; Xu et al., 2021). (Li et al., 2019) proposes to learn a local descriptor and an image-to-class measure to capture the similarity between objects. (Wang et al., 2021) uses a foreground object extractor to exclude the noise from background and synthesize foreground features to remedy the data insufficiency. BSNet (Li et al., 2020c) leverages a bi-similarity module to learn feature maps of diverse characteristics to improve the model's generalization ability. Variational feature disentangling (VFD) (Xu et al., 2021), a data-augmentation method, is complementary to ours. It disentangles the feature representation into intra-class variance and class-discriminating information, and generates additional features for novel classes at test-time. TDM (Lee et al., 2022) applies channel-wise attention to represent different classes with sparse vectors.

**Recognition using Object Parts.** Our method is closely related to recognition based on identifying object components, an approach motivated by how humans learn to recognize object (Biederman, 1987). It draws inspiration from (Ullman et al., 2002), who showed that information maximization with respect to classes of images resulted in visual features eyes, mouth, etc. in facial images and tyres, bumper, windows, etc. in images of cars. Along these lines, Deformable Part Models (DPM) (Felzenszwalb et al., 2009; 2010) proposed to learn object models by composing part features and geometries, and utilize it for object detection. Neural Network models for DPMs were proposed in (Savalle et al., 2014; Girshick et al., 2015). Multi-attention based models, which can be viewed as implicitly incorporating parts, have been proposed (Zheng et al., 2017) in the context of fine-grained recognition problems. Although related, a principle difference is our few-shot setting, where new classes emerge, and we need to generate new object models on-the-fly.

Prior works on FSC have also focused on combining parts, albeit with different notions of the concept. As such, the term part is overloaded, and is unrelated to our notion. DeepEMD (Zhang et al., 2020) focuses in the image-distance metric based on an earth mover's distance between different parts. Here, parts are simply different physical locations in the image and not a compact collection of salient parts for recognition. (Tang et al., 2020) uses salient object parts for recognition, while (Tokmakov et al., 2019) attempts to encode parts into image features. However, both these methods require additional attribute annotations for training, which may be expensive to gather and not always available. (Hao et al., 2019) and (Wu et al., 2021) discover salient object parts and use them for recognition via attention maps similar to our method. They additionally re-weigh their similarity task-adaptively, which is also a feature of our method. We differ in our use of a finite dictionary of templates for learning a compact representation of parts. Also, we use reconstruction as supervision for accurately localizing salient object parts, and impose a meaningful prior on the geometry of parts, which keeps us from degenerate solutions for part locations. For a more detailed comparison with prior related work in fine grained FSC, please refer to the supplementary (Appendix B).

## 3 DEEP OBJECT PARSING

**Parsing Instances.** Each input instance to our method is first parsed using learned templates into a higher level syntax, in the form of parts. While this term, "parts", is overloaded in prior works, our notion of a part is a tuple, consisting of part-location and part-expression at that location.

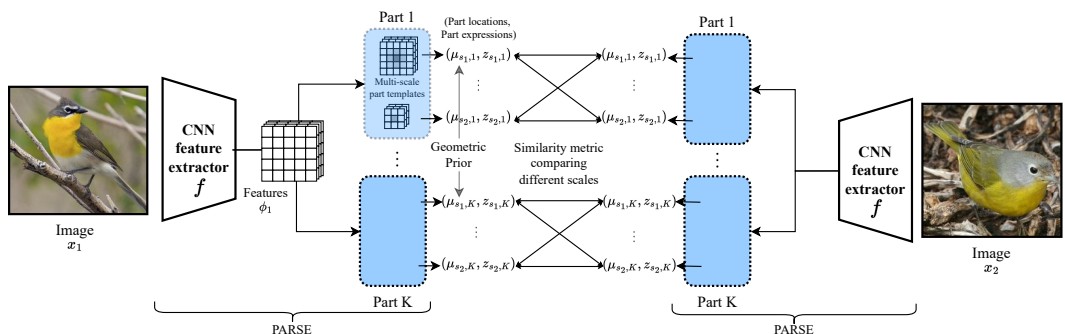

Figure 2: **Deep Object Parsing.** An image $x$ is parsed as a collection of salient parts ($K$ in number). Each part is represented by a 2D location $\mu$ and part expression vector $z$. We notate this operation PARSE and describe it in detail in Alg. 1. In our method, we estimate locations and expressions at multiple scales for each part (hence there are more than one $\mu$ and $z$ per part) and using these, determine image similarity for few-shot recognition. (Best viewed with zoom)

Introducing notation, let $x \in \mathcal{X}$ be an input instance (in our case, an image), $f : \mathcal{X} \to \mathbb{R}^{G \times G \times C}$ a convolutional neural network (CNN) backbone and $\phi = f(x)$ features of $x$, with $C$ channels supported on a 2D $G \times G$ grid. We parse $x$ into $K$ distinct part-locations $\mu_p \in [G] \times [G]$ and part expressions $z_p \in \mathbb{R}^C$ for $p \in [K]$. In our method, we also learn a dictionary of feature-space templates $D_{p,c} \in \mathbb{R}^{s \times s}, p \in [K], c \in [C]$ that are used to represent part features of different instances across different categories.

Given an $s \times s$ mask $M(\mu_p)$ centered at $\mu_p$ (with $s < G$), the learned templates reconstruct part features $\phi$ with the $z_p$ acting as codes : $\phi_{c, M_{\mu_p}} \approx z_{p,c} D_{p,c}$, where the subscript $M_{\mu_p}$ denotes a projection onto the support of $M(\mu)$ (or simply an $s \times s$ window cut-out of $\phi_c$ centered at $\mu_p$). Note that instead of using multiple, we used one dictionary atom per part per channel. While more atoms can reconstruct features better, we found experimentally that they did not benefit few-shot learning performance.

**Part Expression as LASSO Regression.** Given an instance $x$, its feature output, $\phi$, and a candidate part-location, $\mu$, we can estimate sparse part-expression coefficients $z_p(\mu) \in \mathbb{R}^C$ by optimizing the $\ell_1$ regularized reconstruction error, at the location $\mu = \mu_p$ ($\lambda$ being the regularization constant).

$$z_p(\mu) = \arg\min_{\beta} \sum_{c \in C} \|\phi_{c, M(\mu)} - D_{p,c} \beta_c\|^2 + \lambda \|\beta\|_1. \tag{1}$$

*Non-negativity.* Part expressions $z_{p,c}$ signify presence or absence of part templates in the observed feature vectors, and as such can be expected to take on non-negative values. This fact turns out to be useful later for DNN implementation.

**Part Location Estimation.** Note that part expression $z_p$ is a function of location $\mu$, while the part location $\mu_p$ can be estimated by plugging in the optimal part-expressions for each candidate location value, namely,

$$\mu_p = \arg\min_{\mu \in [G] \times [G]} \sum_{c \in C} \|\phi_{c, M(\mu)} - D_{p,c} z_{p,c}(\mu)\|^2 + \lambda \|z_p(\mu)\|_1 \tag{2}$$

This couples the two estimation problems, and is difficult to implement with DNNs, motivating our approach below.

**Feedforward DNNs for Parsing.** To make the proposed approach amenable to DNN implementation, we approximate the solution to Eq. 1 by optimizing the reconstruction error followed by thresholding, namely, we compute $z'_p(\mu) = \arg\min_{\beta} \sum_{c \in C} \|\phi_{c, M(\mu)} - D_{p,c} \beta_c\|^2$, and we threshold the resulting output by deleting entries smaller than $\zeta$: $S_\zeta(u) = u \mathbf{1}_{|u| \geq \zeta}$. This is closely related to thresholding methods employed in LASSO (Hastie et al., 2001).

The quadratic component of the loss allows for an explicit solution, and the solution reduces to template matching per channel, which can further be expressed as a convolution (Gonzalez & Woods,

2008). Using this insight, we derive our estimate of $\mu_p$ as

$$\mu_p = \arg\max_{\mu \in [G] \times [G]} \sum_{c \in C} ((\theta_{p,c} * \phi_c)(\mu) - \lambda_c)^2 \qquad (3)$$

where $*$ is convolution, $\theta_{p,c} = D_{p,c}/\|D_{p,c}\|$, and $\lambda_c = \lambda/2\|D_{p,c}\|$ becomes a channel dependent constant. With the above estimate of $\mu_p$, we get the estimate of $z_p$ as (recall $S_\zeta(u) = u\mathbf{1}_{|u| \geq \zeta}$):

$$z'_{p,c} = \frac{(D_{p,c} * \phi_c)(\mu_p)}{\|D_{p,c}\|^2}; \;\; z_{p,c}(\mu) = S_\zeta(z'_{p,c}) \qquad (4)$$

For a full derivation of the above estimates, please refer to the supplementary materials (App. A).

*Estimates differentiable in parameters.* Since argmax is a non-differentiable function, using Eq. (3) for estimating part-locations does not allow us to use gradient based learning for the parameters of the DNN. We can circumvent this by approximating the argmax as the expectation of a softmax distribution $\nu_p$ over $[G] \times [G]$ with a low temperature $T$.

$$\nu_p(\mu) \triangleq \operatorname{softmax}\left(\frac{1}{T}\sum_{c \in C}((\theta_{p,c} * \phi_c)(\mu) - \lambda_c)^2\right); \quad \mu_p = \mathbb{E}_{\mu \sim \nu_p}\mu \qquad (5)$$

$$z'_{p,c} = \left[\frac{(D_{p,c} * \widehat{\delta}_{\mu_p}) : \phi_c}{\|D_{p,c}\|^2}\right]; \;\; z_{p,c} = S_\zeta(z'_{p,c}) \qquad (6)$$

where $\widehat{\delta}_{\mu_p}$ is a differentiable approximation of a dirac delta centered at $\mu_p$ using a narrow normal distribution and ':' is the double-dot product or the sum of all elements of an element-wise/Hadamard product. Our derivation(App. A) hence leads to very simple expressions, where part-locations $\mu_p$ come from template matching (or convolving the templates) with the CNN features and pooling the product of location indices and $\nu_p$. Part-expressions are then found via a simple convolution and dot product (Eq. (6)).

*Multi-Scale Extension.* We extend our approach to incorporate parsing parts at multiple scales. This is often required because of significant difference in orientation and pose between query and support examples. To do so we simply consider masks $M(\mu)$ and templates $D$ at varying mask sizes $s \in \mathcal{S}$, each leading to independent part location and expression estimates $(\mu_{s,p}, z_{s,p})$ for part $p$. Alg. 1 specifies the parse of an input instance and Fig. 2 shows an overview of object parsing.

---

**Algorithm 1:** PARSE (Object Parsing using DNNs)

1 **Given:** Convolutional backbone $f$, templates $\{D_{s,p,c}\}_{s \in \mathcal{S}, p \in [K], c \in [C]}$, threshold $\zeta$, $\ell_1$ regularization constant $\lambda$, temperature $T$
2 **Input:** Image $x$
3 Compute convolutional features $\phi = f(x)$
4 **for** $p \in [K], s \in \mathcal{S}$ **do**
5      Estimate $\mu_{s,p}$ (through $\nu_{s,p}$) using Eq. (5)
6      Estimate $z_{s,p} = [z_{s,p,c}]_{c \in [C]}$ using Eq. (6)
7 **end for**
8 **Output**: Part locations and expressions $(\{\mu_{s,p}\}_{p \in [K], s \in \mathcal{S}}, \{z_{s,p}\}_{p \in [K], s \in \mathcal{S}})$

---

### 3.1 FEW-SHOT RECOGNITION

At test-time we are given a query instance, $q$, and by way of supervision, $M$ support examples each for $N$ classes, and the goal is to predict the query class label $y^{(q)} \in [N]$. We first run PARSE (Alg. 1) on each of these. $\text{PARSE}(q) = (\{\mu_{s,p}^{(q)}\}, \{z_{s,p}^{(q)}\})$ and for the $i^{th}$ support example of class $y$, $\text{PARSE}(x^{(i,y)}) = (\{\mu_{s,p}^{(i,y)}\}, \{z_{s,p}^{(i,y)}\})$. For comparing query and support examples we need a notion of distance/similarity, which we define next.

*Goodness-of-fit reweighting.* The entropy of the distribution $\nu_{s,p}$ is an important indicator of goodness-of-fit of the dictionary templates (lower entropy meaning a more precise and confident part-location prediction as a result of a better fit). Let $h_{s,p}^{(q)}$ and $h_{s,p}^{(i,y)}$ be the entropies of $\nu_{s,p}^{(q)}$ and

$\nu_{s,p}^{(i,y)}$ respectively. To use these as weights for computing distance (as below), we learn a simple parametric function $\alpha : \mathbb{R}^{M+1} \to \mathbb{R}$.

Additionally, with $z_{s,p}^{(y)} = \frac{1}{M} \sum_{i \in [M]} z_{s,p}^{(i,y)}, s \in \mathcal{S}, p \in [K]$ we represent the mean part expression over all support examples in class $y$. With these, we define the following distance measure between the query example $q$ and the support examples of class $y$.

$$d(q,y) = \sum_{p \in [K]} \sum_{s_1 \in \mathcal{S}} \sum_{s_2 \in \mathcal{S}} \alpha(h_{s_2,p}^{(q)}, [h_{s_1,p}^{(i,y)}]_{i \in [M]}) \left\| z_{s_1,p}^{(y)} - z_{s_2,p}^{(q)} \right\|^2 \quad \text{[expression distance]}$$

$$+ \gamma \sum_{i \in [M]} \sum_{s_1 \in \mathcal{S}} \sum_{s_2 \in \mathcal{S}} \left\| \psi([\mu_{s_1,p}^{(i,y)}]_{p \in [K]}) - \psi([\mu_{s_2,p}^{(q)}]_{p \in [K]}) \right\|^2 \quad \text{[geometric distance]} \quad (7)$$

where $\psi([\mu_{s,p}]_{p \in [K]})$ is a vector of pairwise distances between all part locations at scale $s$, normalized to unit sum. The distance function consists of an expression term, and a geometric term with $\gamma$ acting as a tunable weight to control the proportion of the two. Each term is a sum over all combinations of part scales over query and support. Note that the geometric term simply attempts to find if two polygons with vertices at part locations are similar (i.e. have proportional sides), with the distance being 0 if they are. Finally, the class prediction is made as $\widehat{y}^{(q)} = \arg\min_{y \in [N]} d(q,y)$.

**Training.** We train in episodes following convention. For each episode, we sample $N$ classes at random, and additionally sample support and query examples belonging to these classes from training data (details in Sec. 4). Using a softmax over the negative distance function above as the class distribution of query $q$, we define the cross-entropy loss as

$$\ell_{CE}(q) = -\log \frac{\exp(-d(q, y^{(q)}))}{\sum_{y \in [N]} \exp(-d(q,y))} \quad (8)$$

Additionally, while training, we impose a geometric prior to get diverse instance parts in PARSE by maximizing the Hellinger distance (Everitt, 1998) $\mathbb{H}(\cdot, \cdot)$ between part distributions. The corresponding criterion for minimization is

$$\ell_{div}(x) = -\sum_{s \in \mathcal{S}} \sum_{\substack{p,p' \in [K] \\ p \neq p'}} \mathbb{H}(\nu_{s,p}, \nu_{s,p'}) \quad (9)$$

Alg. 2 outlines the loss computation for a single query $q$. In Line 6, $\eta$ is a tunable parameter controlling the weight of the prior. In each episode, we use an average of the loss output over multiple query examples, which results in an end-to-end differentiable criterion in all trainable parameters, allowing us to optimize using gradient descent.

---

**Algorithm 2:** Training loss for Few-shot recognition with DOP (single episode, single query)

---

1 **Given:** requirements for Alg. 1 PARSE, weighting function $\alpha$, tunable parameter $\gamma$

2 **Input:** Query example with ground truth label $q, y^{(q)} \in \mathcal{X} \times [N]$. Support examples
  $I = \bigcup_{y \in [N]} \left[ I_y = \{x^{(i,y)}\}_{i \in [M]} \right]$

3 **Trainable parameters:** Convolutional backbone $f$, part templates $\{D_{s,p,c}\}_{s \in \mathcal{S}, p \in [K], c \in [C]}$, weighting function $\alpha$

4 Compute parses PARSE$(q) = (\{\mu_{s,p}^{(q)}\}, \{z_{s,p}^{(q)}\})$,
  PARSE$(x^{(i,y)}) = (\{\mu_{s,p}^{(i,y)}\}, \{z_{s,p}^{(i,y)}\}); s \in \mathcal{S}, p \in [K]$

5 Compute distances $d(q,y)$ for $y \in [N]$ using Eq. (7)

6 **Output**: loss $\ell(q) = \ell_{CE}(q) + \eta \frac{1}{|I|+1} \sum_{x \in I \cup \{q\}} \ell_{div}(x)$ using Eqs. (8) and (9)

---

## 3.2 IMPLEMENTATION DETAILS

We use two Resnet (He et al., 2016) backbones (Resnet-12 and Resnet-18) as feature extractors. We use the Resnet-12 model used in prior works (Lee et al., 2019), which has more output channels and more parameters (12.4M) compared to Resnet-18 (11.2M). The input image is resized to $84 \times 84$ for Resnet-12 and $224 \times 224$ for Resnet-18. In the output features, for Resnet-12, $C = 640$ and $G = 5$, and for Resnet-18, $C = 512$ and $G = 7$. The number of parts $K$ is set to 4 for most experiments (see Sec. 4.2 for a experiments with different number of parts). There are three scales

$\mathcal{S} = \{1, 3, 5\}$ considered for each part. The temperature $T$ in Eq. (5) is set to 0.01 and the threshold $\zeta$ in Eq. (6) is set to 0.05. $\gamma$ is set to 0.01. For $\alpha$ we used a linear layer, and subsequently normalized $[\alpha(h_{s_1,p}^{(q)}, [h_{s_2,p}^{(i,y)}]_{i \in [M]})]_{p \in [K]}$ (a K-dimensional vector; see Eq. (7)) with a softmax.

## 4 EXPERIMENTS

### 4.1 FINE-GRAINED FEW-SHOT CLASSIFICATION

We compare DOP on four fine-grained datasets: Caltech-UCSD-Birds (CUB) (Wah et al., 2011), Stanford-Dog (Dog) (Khosla et al., 2011) Stanford-Car (Car) (Krause et al., 2013) and Aircraft (Maji et al., 2013) against state-of-the-art methods.

**Caltech-UCSD-Birds (CUB)** (Wah et al., 2011) is a fine-grained classification dataset with 11,788 images of 200 bird species. Following convention(Hilliard et al., 2018), the 200 classes are randomly split into 100 base, 50 validation and 50 test classes.

**Aircraft** contains 100 classes of aircrafts and 10,000 images in total. Following recent benchmark (Lee et al., 2022; Wertheimer et al., 2021), we processes all images based on bounding box. And the 100 classes are split into 50, 25 and 25 classes for training, validation and test.

**Stanford-Dog/Car** (Khosla et al., 2011; Krause et al., 2013) are two datasets for fine-grained classification. Dog contains 120 dog breeds with a total number of 20,580 images, while Car consists of 16,185 images from 196 different car models. For few-shot learning evaluation, we follow the benchmark protocol proposed in (Li et al., 2019). Specifically, 120 classes of Dog are split into 70, 20, and 30 classes, for training, validation, and test, respectively. Similarly, Car is split into 130 train, 17 validation and 49 test classes.

**Experiments Setup.** We conducted 5-way (5 classes episode) 1-shot and 5-way 5-shot classification tasks on all datasets. Following the episodic evaluation protocol in (Vinyals et al., 2016), at test time, we sample 600 episodes and report the averaged Top-1 accuracy. In each episode, 5 classes from the test set are randomly selected. 1 or 5 samples for each class are sampled as support data, and another 15 examples are sampled for each class as the query data. The model is trained on train split and the validation split is used to select the hyper-parameters. We compare our DOP to state-of-the-art FSC and fine-grained FSC methods. More details in the supplementary (App. B).

**Training Details.** Our model is trained with 10,000 episodes on CUB and 30,000 episodes on Stanford-Dog/Car for experiments with both ResNet12 and ResNet18. In each episode, we randomly select 10 classes and sample 5 and 10 samples as support and query data. The weight on the geometric prior $\eta$ is set to 1.0 on CUB and 0.1 on Stanford-Dog/Car, respectively. We train from scratch with Adam optimizer (Kingma & Ba, 2014). The learning rate starts from 5e-4 on CUB and 1e-3 on Stanford-Car/Dog, and decays to 0.1x every 3,000 episodes on CUB and 9,000 episodes on

Table 1: Few-shot classification accuracy in % on Aircraft dataset. Unless specified, methods use ResNet-12 backbone.

| Methods | 1-shot | 5-shot |
|---|---|---|
| ProtoNet(Snell et al., 2017) | 67.28 | 83.21 |
| DSN(Simon et al., 2020) | 70.23 | 83.03 |
| CTX(Doersch et al., 2020) | 71.57 | 79.31 |
| FRN(Wertheimer et al., 2021) | 69.58 | 83.98 |
| HelixFormer(Zhang et al., 2022) | 74.01 | 83.11 |
| TDM(Lee et al., 2022) | 71.57 | 84.77 |
| DOP(ResNet18) | **84.26** | **93.41** |
| DOP(ResNet12) | **85.50** | **94.35** |

Dog/Car. On CUB, objects are cropped using the annotated bounding box before resizing to the input size. On Stanford-Car/Dog, we use the resized raw image as the input. We employed standard data augmentations, including horizontal flip and perspective distortion, to the input images.

**Results.** Our results along with comparisons against state-of-the-art on CUB, Aircraft and Stanford-Dog/Car are reported in Tabs. 1, 2 and 3, respectively.

**DOP is competitive with or outperforms recent works on fine-grained FSC.** On CUB (Tab. 2), DOP outperforms all compared approaches with 83.39% 1-shot accuracy and 93.01% for 5-shot with Resnet-12 backbone. Same is the case for Aircraft (Tab. 1), where DOP outperforms a recent method in TDM (Lee et al., 2022) by a large margin using ResNet-12 backbone, performing at 85.50% 1-shot and 94.35% 5-shot accuracy. While on Stanford-Car (Tab. 3), we outperform compared approaches by 3.06% and 1.85% on 1-shot and 5-shot, respectively, on Stanford-Dog, we outperform

Table 2: Few-shot accuracy in % on CUB compared to most recent state-of-the-art methods (along with 95% confidence intervals). If not specified, the results are those reported in the original paper. †: results are obtained by running the public implementation released by authors using ResNet18 backbone. A more comprehensive comparison to all related methods can be found in App. C.

| Methods | Backbone | 1-shot | 5-shot |
|---|---|---|---|
| ProtoNet(Snell et al., 2017) | ResNet18 | 71.88±0.91 | 87.42±0.48 |
| DeepEMD(Zhang et al., 2020) | ResNet12 | 75.65±0.83 | 88.69±0.50 |
| FOT(Wang et al., 2021) | ResNet18 | 72.56±0.77 | 87.22±0.46 |
| VFD (Xu et al., 2021) | ResNet12 | 79.12±0.83 | 91.48±0.39 |
| FRN(Wertheimer et al., 2021) | ResNet12 | 83.16 | 92.59 |
| RENet(Kang et al., 2021) | ResNet12 | 79.49±0.44 | 91.11±0.24 |
| TOAN(Huang et al., 2021) | ResNet12 | 67.17± 0.81 | 82.09±0.56 |
| RAP(Hong et al., 2021) | ResNet18 | **83.59±0.18** | 90.77±0.10 |
| TDM(Lee et al., 2022) | ResNet12 | 83.36 | 92.80 |
| HelixFormer(Zhang et al., 2022) | ResNet12 | 81.66±0.30 | 91.83±0.17 |
| DOP | ResNet18 | 82.62±0.65 | **92.61±0.38** |
| DOP | ResNet12 | 83.39±0.82 | **93.01±0.43** |

all methods but VFD. We note here that VFD generates additional features at test-time for novel classes, and is as such, complementary to DOP.

Table 3: Few-shot classification accuracy in % on Stanford-Car/Dog benchmarks (along with 95% confidence intervals). †: results are obtained by running the codes released by authors using ResNet18 backbone.

| Methods | Backbones | Car | | Dog | |
|---|---|---|---|---|---|
| | | 1-shot | 5-shot | 1-shot | 5-shot |
| ProtoNet†(Snell et al., 2017) | ResNet18 | 60.67±0.87 | 75.56±0.45 | 61.06±0.67 | 74.31±0.51 |
| MetaOptNet†(Lee et al., 2019) | ResNet18 | 60.56±0.78 | 76.35±0.52 | 65.48±0.56 | 79.39±0.43 |
| BSNet(Li et al., 2020c) | ResNet12 | 60.36±0.98 | 85.28±0.64 | 69.09±0.90 | 82.45±0.58 |
| VFD(Xu et al., 2021) | ResNet12 | - | - | **76.24±0.87** | **88.00±0.47** |
| TOAN(Huang et al., 2021) | ResNet12 | 76.62±0.70 | 89.57±0.40 | 51.83±0.80 | 69.83±0.66 |
| HelixFormer(Zhang et al., 2022) | ResNet12 | 79.40±0.43 | 92.26±0.15 | 65.92±0.49 | 80.65±0.36 |
| DOP | ResNet18 | **81.41±0.71** | **93.48±0.38** | 70.56±0.75 | 84.75±0.41 |
| DOP | ResNet12 | **81.83±0.78** | **93.84±0.45** | 70.10±0.79 | 85.12±0.55 |

## 4.2 ANALYSES

**From Prototypical Networks to Deep Object Parsing.** As an overview, DOP combines object parsing, dictionaries at multiple template sizes, use of part geometry for distance computation and instance-dependent distance reweighting based on goodness-of-fit. These can be seen as methodological developments over ProtoNet (Snell et al., 2017), a simple CNN feature-space distance based few shot classification approach.

*ProtoNet to single part DOP.* Let us consider a simplified DOP method with a single part ($K = 1$) and parsing done at a single scale (template size 5) $\mathcal{S} = \{5\}$. There is thus one template $D \in \mathbb{R}^{5 \times 5 \times C}$ consisting of learned parameters. In estimating the part location (Eq. (5)), we convolve the template over the CNN features, and perform some additional operations (Eq. (6)) with no additional learnable parameters to find part expressions. A ProtoNet with the same number of learnable parameters can use $D$ as a final convolutional layer and perform global pooling over its outputs. On 5-way 5-shot classification on CUB this ProtoNet has accuracy 88.38%, while the simplified DOP has accuracy 90.56%. Thus, this improvement in performance can be attributed to learning a template shareable across parts that can be used for reconstructing features (using the part expressions $z$). This reconstruction objective allows the part expressions $z$ to have less noise and

thus lower intra-class variance. Next, we analyze what happens when we parse objects into multiple parts and at multiple scales.

*Using multiple parts.* Tab. 4 shows the effect of adding more parts on 5-way 5-shot accuracy on CUB. We see more parts up to a certain point ($K = 4$) allows DOP to learn better representations consequently improving performance, but with even more parts performance drops as the model can start learning irrelevant or background signatures.

*Using templates at multiple scales.* In Tab. 5 using the Stanford-Dog dataset, we studied the effect of parsing parts at multiple scales. Learning dictionaries at multiple scales improves performance, since this allows DOP to parse the object parts even when their scale may vary (due to different positions and orientations relative to the camera).

Table 4: Effect of using different number of parts on 5-way 5-shot accuracy on CUB.

| Num parts | 1 | 3 | 4 | 5 | 6 |
|---|---|---|---|---|---|
| Accuracy | 90.56 | 92.10 | **92.61** | 92.21 | 92.06 |

Table 5: Effect of using templates at different scales on 5-way 5-shot accuracy on Dog.

| Scales | [3] | [5] | [3,5] | [1,3,5] |
|---|---|---|---|---|
| Accuracy | 81.56 | 81.38 | 83.04 | **84.75** |

In the supplementary, we perform additional analysis ablating the use of part geometries for distance computation and the use instance dependent reweighting $\alpha$.

**What parts does DOP detect?** We visualize the locations $\mu_p$ learned by DOP in Fig. 3. DOP is able to detect consistent parts for the same task and often finds semantically meaningful parts like head and torso/breast in birds and dogs and wheels and doors/windows on cars. Fig. 3 also shows some failure cases of DOP, where it might fail to locate parts on the object if similar visual signatures appear in the background. A visualization of part templates learned and part expressions can be found in the supplementary (App. D).

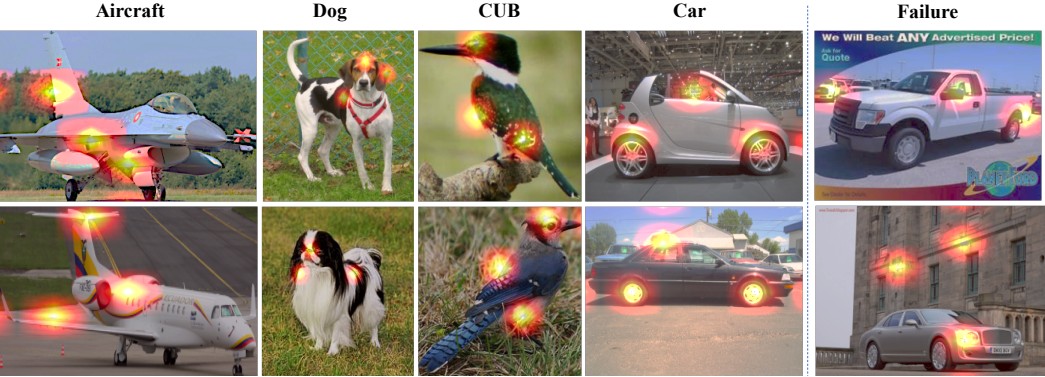

Figure 3: Exemplar part locations learned by DOP when $K = 3$. From left to right: Aircraft, CUB, Dog, Car, and failure cases. DOP can fail and locate parts on the background if it has visual signatures similar to an object.

## 5  CONCLUSIONS

We presented DOP, a deep object-parsing method for fine-grained few-shot recognition. Our fundamental concept is that, while different object classes exhibit novel visual appearance, at a sufficiently small scale, visual patterns are duplicated. Hence, by leveraging training data to learn a dictionary of templates distributed across different relative locations, an object can be recognized simply by identifying which of the templates in the dictionary are expressed, and how these patterns are geometrically distributed. We build a statistical model for parsing that takes the output of a convolutional backbone as input to produce a parsed output. We then post-hoc learn to re-weight query and support instances to identify the best matching class, and as such this procedure allows for mitigating visual distortions. Our proposed method is an end-to-end deep neural network training method, and we show that our performance is not only competitive but also the outputs generated are interpretable.

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
