# OpenReview forum: "Fine-grained Few-shot Recognition by Deep Object Parsing"
_ICLR.cc/2023/Conference — Submitted to ICLR 2023_

### Official Review · Reviewer_47uq · 2022-10-24

**Confidence:** 4
**Correctness:** 4
**Technical Novelty And Significance:** 3
**Empirical Novelty And Significance:** 2
**Recommendation:** 6

**Clarity, Quality, Novelty And Reproducibility:**

The paper is well structured and clearly written. Still the definition of the statistical method is hard to follow.

Looking at fig.3 it seems that the statistical model extracts too local features. How robust could be the model? For example in the last column the truck and the car are characterised with only 2 and 1 features correspondingly.

**Strength And Weaknesses:**

The strengths of the paper are:
1. The main assumption is logic and intuitive.
2. The statistical method is straightforward and technically sound.
3. The method also can facilitate its interpretability and explainability since express explicitly the visual dictionary and the relevant parts of the object.
4. The paper is well organised.
5. Validation is done on 4 public datasets.
6. Results are promising.
7. Ablation study is presented.

Weaknesses:
1. State of the art should be completed and the method should be compared to the following works:
- Huang, S., Wang, X., & Tao, D. (2021). Stochastic partial swap: Enhanced model generalization and interpretability for fine-grained recognition. In Proceedings of the IEEE/CVF International Conference on Computer Vision (pp. 620-629).
- Baffour, A. A., Qin, Z., Wang, Y., Qin, Z., & Choo, K. K. R. (2021). Spatial self-attention network with self-attention distillation for fine-grained image recognition. Journal of Visual Communication and Image Representation, 81, 103368.
- Zhu, H., & Koniusz, P. (2022). EASE: Unsupervised Discriminant Subspace Learning for Transductive Few-Shot Learning. In Proceedings of the IEEE/CVF Conference on Computer Vision and Pattern Recognition (pp. 9078-9088).
- Lazarou, M., Stathaki, T., & Avrithis, Y. (2021). Iterative label cleaning for transductive and semi-supervised few-shot learning. In Proceedings of the IEEE/CVF International Conference on Computer Vision (pp. 8751-8760).
- Hong, J., Fang, P., Li, W., Zhang, T., Simon, C., Harandi, M., & Petersson, L. (2021). Reinforced attention for few-shot learning and beyond. In Proceedings of the IEEE/CVF Conference on Computer Vision and Pattern Recognition (pp. 913-923).
- Fei, N., Gao, Y., Lu, Z., & Xiang, T. (2021). Z-score normalization, hubness, and few-shot learning. In Proceedings of the IEEE/CVF International Conference on Computer Vision (pp. 142-151), etc.
2. Looking at the papers mentioned above, it is not clear that the final results are outperforming the state of the art as claimed in the paper.
3.  Also, it is not clear why the authors didn't use more popular datasets for few-shot learning like miniImageNet, tieredImageNet, CIFAR-FS and FC100.
4. Curiously, the paper defines the contributions as a method that outperforms the Stanford-Car few shot fine-grained learning as well as the ablation study and the visualisation aspect of the method. It is recommended to define better the theoretical contributions.


**Summary Of The Paper:**

This paper addresses the problem of few-shot fine-grained recognition. The work is based on the generic assumption that while different object classes exhibit novel appearance, visual patterns are duplicated. Thus, the proposed method is forced to learn a dictionary of templates distributed across different relative locations. As a result an object can be recognised by identifying its corresponding parts in term of visual templates and their relation.  The whole process is expressed as a statistical model for parsing. Given a query image, the model re-weights the query and supports instances to find out the best matching class. The method is validated on 4 public datasets giving the results in terms of 5-way top-1 and 5-way top-5 accuracy.

**Summary Of The Review:**

The assumption is intuitive, the statistical model is technically sound but it is not clear that the method outperforms the state of the art and in which of the datasets (as stated the authors mentioned that only the Stanford-Car dataset is improved).

---

> ### Author Response · Authors · 2022-11-19
> **Initial Response**
>
> Thank you very much for your constructive comments and questions.
>
> > Updating state of the art (referring to the 6 papers mentioned as [a]...[f]).
>
> We added RAP[e] and IEPT+ZN[f] to our comparisons on the CUB benchmark (Table 1 in the revised paper; also in the table below). While RAP is marginally better on 5-way 1-shot classification, it underperforms DOP and other more recent methods on 5-way 5-shot classification.
> [a] and [b] are fine-grained recognition models while DOP as a method is shown to be effective for few-shot fine-grained recognition.
> [c] and [d] are methods that transductively solve few-shot classification, meaning they can use more information in the form of unlabelled test data and label all of them together. This is a different scenario than what DOP operates in, where it labels each test image individually.
>
> | Methods | Backbone | CUB 1-shot | CUB 5-shot|
> |-------------|:-------------:|:---------------:|:---------------:|
> | RAP [e] | ResNet18 | **83.59±0.18** | 90.77±0.10 |
> | IEPT+ZN [f] | Conv4-64 | 73.54 ± 0.48 | 87.82 ± 0.30 |
> | DOP | ResNet18 | 82.62±0.65 | **92.61±0.83** |
> | DOP | ResNet12 | 83.39±0.82 | **93.01±0.43** |
>
> > Comparison on few-shot learning benchmarks.
>
> DOP as a method is proposed for better fine-grained few-shot recognition. It is built on the assumption that in these problems, common parts across different instances can be represented using shared dictionaries of templates. Hence, we do not include comparisons on few-shot recognition benchmarks that are not based on fine-grained datasets like mini-Imagenet, tiered-Imagenet, CIFAR-FS and FC-100.
>
> > "Contributions" paragraph of introduction only mentions Stanford-Cars.
>
> Thank you for pointing it out. Mentioning only Stanford-Cars was an error in writing. We have updated our contributions paragraph to fix this and also included your suggestion regarding the derivation of our method leading to a final simplified form.

---

### Official Review · Reviewer_cctR · 2022-10-24

**Confidence:** 4
**Correctness:** 3
**Technical Novelty And Significance:** 3
**Empirical Novelty And Significance:** 2
**Recommendation:** 3

**Clarity, Quality, Novelty And Reproducibility:**

The clarity of the paper is not good, the novelty of the paper is non-trivial, and the reviewer is not very confident in reproducing the results of the DOP since there omit lots of implementation and training details.

**Strength And Weaknesses:**

Strength:
1. The idea of learning a dictionary of parts for fine-grained few-shot image classification is correct and reasonable.
2. The experimental results reported validate the effectiveness of the proposed DOP, and DOP achieved state-of-the-art performance on CUB, Car, and Aircraft datasets.

Weaknesses:
1. The motivation of the paper is somewhat weak. The connections between fine-grained few-shot image classification and deep object parsing are not well illustrated, especially for the intra-class variations that will enlarge under the few-shot settings.
2. Some formulations in the paper are hard to follow. For example, on page 4, "The location corresponds to an $s \times s$ attention mask'', "$D_{p,c} \in R^{s \times s}$ '', what's the attention mask used for? what's the connection between the $G \times G$ 2D feature map slice and the $s \times s$ attention mask? "$ M(\mu) $ centered round the location $\mu$", what's the function of $M(\mu)$?
3. As presented in the paper, the DOP needs to use DNNs to parse the input objects and generate the aligned-structured features. Is the DNNs in Algorithm 1 a pre-trained model or not? How to obtain these DNNs? If it is a pre-trained one, does the pre-trained dataset contains testing images or directly grab from pre-trained models on ImageNet?
4. As the DOP consists of two steps, it imports additional operations of object parsing using DNNs. The model complexity, total parameters and running time comparisons with other single models should be investigated.
5. In the experimental comparisons, the compared methods seem to vary a lot. On CUB, there are 15 baseline models. On Aircraft, Car, and Dog, there are only 5 to 7 baseline models. Moreover, equipped with complex deep object parsing, the DOP model achieves slight improvement on CUB, with only 0.03 and 0.21 boosts on 1-shot and 5-shot image classification over TDM. In Table 3, the accuracy of DOP lags behind the VFD by a wide margin on Dog, and more explanations or clarifications are encouraged.
6. There exist previous works [C1,C2,C3] that proposed aligning and learning the fine-grained features in an end-to-end way, which are also inspired by the template learning in fine-grained image classification. Comprehensive comparisons are suggested.
7. The visualization of the learned templated parts seems inaccurate in some correct cases in Figure 3. For example, the learned parts of Dog are not matched. In the top image, the locations are head, ear and body, while in the bottom image, the locations are nose, body and body.

C1. Local spatial alignment network for few-shot learning
C2. TOAN: Target-Oriented Alignment Network for Fine-Grained Image Categorization with Few Labeled Samples
C3. Learning Cross-Image Object Semantic Relation in Transformer for Few-Shot Fine-Grained Image Classification

**Summary Of The Paper:**

This paper studied fine-grained image recognition under few-shot settings. The authors proposed to learn a dictionary of parts using object parsing first and then train a few-shot learner based on the parsed features. Learning a structured aligned representation is the right way for fine-grained few-shot learning. The high intra-class variations enlarge with the limited labeled samples in each category, which is more challenging than traditional fine-grained image recognition. It is interesting that the paper proposes to learn parsed object features from the perspective of dictionary learning.

**Summary Of The Review:**

This paper proposed to solve the fine-grained few-shot image classification using deep object parsing. The idea is reasonable and interesting. However, the motivation of the paper is not well illustrated, and many other concerns exist, as discussed in the Strength And Weaknesses section. Therefore, the reviewer leans to reject the submission and encourages the authors to improve the paper and resubmit to other venues.

---

> ### Author Response · Authors · 2022-11-19
> **Initial response**
>
> Thank you very much for your constructive comments and questions.
>
> > Link between motivation and fine-grained few shot classification.
>
> In the revised introduction, we made this more explicit. We mention that DOP assumes "Duplication of Traits" which occurs in fine-grained classification dataset. Secondly, under "Parsing", we mention that parsed representations using shared dictionaries of templates have low noise, which is helpful in high intra-class variance scenarios like few-shot learning.
>
> > What is $M(\mu)$ used for?
>
> The $s \times s$ mask $M(\mu_p)$ simply allows to get an $s \times s$ window cut-out of $\phi_{c}$ centered at $\mu_p$. We rewrote the first 3 paragraphs of Section 3, in part, to make this more clear, and included the following line in the beginning of 3$^{rd}$ paragraph.
>
> "Given an $s \times s$ mask $M(\mu_p)$ centered at $\mu_p$ (with $s < G$), the learned templates reconstruct part features $\phi$ with the $z_p$ acting as codes : $\phi_{c, M_{\mu_p}} \approx z_{p, c} D_{p, c}$, where the subscript $M_{\mu_p}$ denotes a projection onto the support of $M(\mu)$ (or simply an $s \times s$ window cut-out of $\phi_{c}$ centered at $\mu_p$)."
>
> > Is the DNN pre-trained?
>
> The DNN used is not pre-trained, they are randomly initialized before training. In Sec 3, we derived differentiable estimates for part-locations $\mu$ and expressions $z$. The DNN can be thus be trained along with the templates in an end-to-end fashion.
>
> > How many additional parameters are used and what is the additional latency?
>
> While we did not make a clear demarcation in the paper, we assume the question means
> - Step-1 : DOP extracts CNN features $\phi$
> - Step-2 : Then via template matching gets part locations $\mu_p$ and expressions $z_p$.
> The 2$^{nd}$ step here is a much smaller operation compared to the computations in the rest of the CNN. It involves only the templates as the additional parameters which are 0.02M in number or only 0.17\% of the number of parameters in the ResNet-18 backbone. As a comparison, in a recent state of the art method TDM [a], 0.07M additional parameters are used.
> Finally the feed-forward latency added by Step 2 is ~13\% (147 ms for a Resnet-18 backbone and 19 ms for Step 2 computation). The above calculations use a DOP model with 4 parts parsed at scales {1,3,5}
>
> [a] Task discrepancy maximization for fine-grained few-shot classification.
>
> > DOP performance compared to TDM, VFD.
>
> We find that DOP is competitive with or outperforms state-of-the-art methods on multiple datasets. While DOP outperforms TDM by a small margin on CUB, it performs better by a much bigger margin on the Aircraft dataset. This is possibly due to the fact that object geometries are more consistent across instances in the Aircraft dataset.
>
> In comparison to VFD, using the reported results, DOP performs better on CUB, while it is worse on the Stanford-Dog benchmark. We note here that we attempted reproducing the results of VFD from their public implementation but could not.
>
> Reported results : (CUB : 91.48), (Dog : 88.00)\
> Reproduced : (CUB : 87.34), (Dog : 77.68)
>
> We have reached out to the authors of VFD regarding the reproducibility of their results but have not heard back yet. While on CUB, the difference between reproduced and reported numbers is relatively small (while still significant), the much larger difference on the Stanford-Dog dataset makes the reported number seem unreliable.

---

> > ### Author Response · Authors · 2022-11-19
> > **Initial response [part 2]**
> >
> > > Comparison with other methods along alignment and template matching.
> >
> > Thank you for the pointers, we have added the methods mentioned in Tables 1, 2 and 3 of the revised paper.
> >
> > We compare and contrast the alignment and template matching aspects of these methods here:
> > C1 compares local features which is similar to our comparison of parts. However, their comparison is done at all pairs of part locations and there is no notion of salient parts, as opposed to our approach. C2 refers to templates as category level features and attempts to use a query as a template of its own category. This notion of templates differs from our approach where templates are atoms of a dictionary and learned to represent features for parts across all instances. C3 uses a cross-attention mechanism across query and support which is a modification of self-attention layers from transformer architectures. While this may be similar to C2 and by transitivity to DOP in using features at all 2D locations of CNN output map in an attention layer, there does not seem be more meaningful relations between the methods.
> >
> > In the following table, we compare [C1, C2, C3] with DOP. In terms of FSC accuracy DOP performs better than all three of them.
> >
> > | Methods | Backbone | CUB 1-shot | CUB 5-shot| Car 1-shot | Car 5-shot | Dog 1-shot | Dog 5-shot |
> > |-------------|:-------------:|:---------------:|:---------------:|:---------------:|:---------------:|:---------------:|:---------------:|
> > | TOAN[C2] | ResNet12 | 67.17±0.81 | 92.09±0.56 | 76.62±0.70 | 89.57±0.40 | 51.83±0.80 | 69.83±0.66 |
> > | HelixFormer[C3] | ResNet12 | 81.66±0.30 | 91.83±0.17 | 79.40±0.43 | 92.26±0.15 | 65.92±0.49 | 80.65±0.36 |
> > | LSANet[C1] | Conv-64F | 67.75 | 82.76 | 68.65 | 87.23 | 55.85 | 71.18 |
> > | DOP | ResNet18 | **82.62±0.65** | **92.61±0.83** | **81.41±0.71** | **93.48±0.38** | **70.56±0.75** | **84.75±0.41** |
> > | DOP | ResNet12 | **83.39±0.82** | **93.01±0.43** | **81.83±0.78** | **93.84±0.45** | **70.10±0.79** | **85.12±0.55** |
> >
> >
> > > Mis-located part in dog (Fig. 3).
> >
> > Since part-locations are positions in the feature space which is smaller (typically 7x7 in size) compared to an image (typically 224x224 in size), upscaling the part-location to a location in the image can be inaccurate by a few pixels. Thus, while it seems in Fig 3, the second dog has a 2nd part detected on its body, it is only a few pixels away from the left ear, which is likely the part detected.
> >
> > > Implementation.
> >
> > We have attached a zip of our repository in the revised supplementary materials, with specific instructions in the README to reproduce our results.

---

> > > ### Comment · Reviewer_cctR · 2022-11-24
> > > **Response**
> > >
> > > Thanks for your feedback! Some of my concerns are addressed. However, the template-parts mismatched problem is not well addressed. As the main contribution of the paper is the dictionary learning of the template parts, and the new revision claimed that the DOP is inspired by the "Duplication of Traits", which means the dictionary template of the Fine-grained classes should be highly similar. The quality of the learned templates is thus crucial to DOP. However, the qualitative and quantitative studies of the learned templates are missing, and the mismatched problem indicates the ineffectiveness of the templates-learning mechanism. To this end, the reviewer will not change the recommendation of rejection.

---

> > > > ### Author Response · Authors · 2022-12-05
> > > > **Response on Duplication of Traits**
> > > >
> > > > Thanks for your kind comments and interesting question about the Duplication of Traits.
> > > >
> > > > We updated Figure 2 in the supplementary material showing a quantitative example of the Dog dataset. It demonstrates template coefficients for images of the same class are similar (Top 2 rows). Visually-similar classes (Boston Terrier and Staffordshire Terrier) share some of the same activated templates, while visually distinct classes (Golden Retriever and Other Terriers) differ in some of their selection of active templates. A single channel of the template dictionary may represent some common texture, shapes and hidden characteristics. The particular collection of templates uniquely determines an object part belonging to a particular class.
> > > >
> > > > This example validates our proposition about the Duplication of Traits. In fact, this Figure is one of the common examples of our method. We will show more qualitative results across different datasets in the next version of supplementary material.

---

### Official Review · Reviewer_SUHw · 2022-10-27

**Confidence:** 3
**Correctness:** 2
**Technical Novelty And Significance:** 2
**Empirical Novelty And Significance:** 2
**Recommendation:** 5

**Clarity, Quality, Novelty And Reproducibility:**

Clarity: Good. The paper is easy to follow.
Quality: Good.
Novelty: Borderline.
Reproducibility: Borderline.

**Strength And Weaknesses:**

Strength:
- The method is end-to-end trainable.
- An object instance is represented as a collection of parts with locations, which is somewhat interpretable.
- The method achieves SoTA performance on some datasets.

Weaknesses:
- The novelty is on the weak side since there are previous methods that also parse objects into different salient parts for object recognition [2] [3]. The introduction doesn't discuss these methods and it's hard to clearly see the main contribution of the work. The authors write the following at the end of the related work:
"We differ in our use of a finite dictionary of templates used to learn a compact representation of parts. Also, we use reconstruction as supervision for accurately localizing salient object parts, and impose a meaningful prior on the geometry of parts, which keeps us from degenerate solutions for part locations." From there, I see there are two main points: - a finite dictionary of templates and - using feature reconstruction. The justification here is weak and there is no analysis of these two points.
[2] Task-aware part mining network for few-shot learning
[3]  Collect and select: Semantic alignment metric learning for few-shot learning

- The motivation is not very convincing to me. I think a CNN automatically parses object instances into different parts [1] where each feature might respond to a different discriminative object part. In fact, here the authors also use a pre-trained network. Thus the main contribution here is to explicitly represent them as a fixed collection of K parts rather than directly using the implicit collection of parts learned from the deep network. I am not entirely sure why this representation is more robust than just using the CNN features. Does it come from the explicit locations of parts or is it because the method can extract the top K most informative and discriminative parts? In general, I think I didn't get the answer to "why does this method work?" from a high-level perspective. The connection between motivation and fine-grained few-shot learning is also not very obvious. Should this method work also on the standard setting of fine-grained classification?

[1] Visualizing and Understanding Convolutional Networks

- Parsing objects into a fixed set of parts seems very challenging and requires several technical components involved, e.g., setting the right value of K, using multiple scales, or instance-dependent re-weighting. These components only address the challenges specifically posed in this particular modeling. The solution is quite heavily engineered.



**Summary Of The Paper:**

The authors propose a method for fine-grained few-shot recognition by finding a dictionary of K parts to represent objects. The dictionary is shared across all instances and categories. An object is represented by where are these K parts and a set of templates that reconstruct the part features. The method achieves competitive performance compared to SoTA methods.

**Summary Of The Review:**

My main concern is the novelty and unclear significance of the overall contribution.

---

> ### Author Response · Authors · 2022-11-19
> **Initial response**
>
> Thank you very much for your constructive comments and questions.
>
> > Analysis of DOP components.
>
> To help with a high-level understanding of why DOP performs well, we updated the analysis section (Sec 4.2 in the revision) to methodologically build DOP over ProtoNet, at each stage comparing performances (which includes a new experiment with a simple 1-part DOP). DOP builds in combining the 4 components : object parsing, dictionaries at multiple template sizes, use of part geometry for distance computation and instance-dependent distance reweighting based on goodness-of-fit. The analysis for ablating each has been included, the first two in Sec 4.2 and the last two in the supplementary Table 2 (App. D) (previously Table 4 in the main paper).
>
> > Comparison with other part based methods?
>
> In the supplementary (App. B), we presented a detailed comparison of how our work differs from recent methods using parts, reweighting or reconstruction criteria. We have added a pointer to this from "Related Work" in the revision. As mentioned in the review, DOP differs from both [2] and [3] in the use of learning dictionaries of templates, with a reconstruction target and using part geometries for comparing instances.
>
> > Is DOP backbone pre-trained?
>
> No. The backbone is randomly initialized before training and trained along with the part templates.
>
> > Link between motivation and fine-grained few shot classification.
>
> In the revised introduction, we made this more explicit. We mention that DOP assumes "Duplication of Traits" which occurs in fine-grained classification dataset. Secondly, under "Parsing", we mention that parsed representations using shared dictionaries of templates have low noise, which is helpful in high intra-class variance scenarios like few-shot learning.
>
> > DOP is heavily engineered.
>
> While the points made regarding DOP using multiple components are fair, we also note that DOP is designed for effectiveness in fine-grained few-shot classification, and each component has specific motivations and benefits (as seen in Sec 4.2).

---

### Official Review · Reviewer_qRQa · 2022-11-03

**Confidence:** 3
**Correctness:** 4
**Technical Novelty And Significance:** 3
**Empirical Novelty And Significance:** Not applicable
**Recommendation:** 6

**Clarity, Quality, Novelty And Reproducibility:**

Generally speaking, the paper is in good quality and the presentation is quite clear. It's somewhat novel that the paper integrates recent effective deep modules (e.g. attention and feature map reconstruction) into traditional part-based models.

The paper hasn't released their code yet, so it's a bit hard for others to reproduce their work.





**Strength And Weaknesses:**

Strength:
* The proposed model is technically sound and effective, and it's end-to-end trainable. It successfully combines the advantages of good interpretability in traditional part-based models and nice performance improvement brought by deep networks and modules.
* Generally speaking, the presentation is clear and the experiments are sound.

Weakness:
* The math formulas in Sec 3 are a bit difficult to understand. It's better to explain the variables more clearly immediately before each formula.
* Possible typos in math formulas:
  * In the 3rd paragraph of sec 3.1, z_{s,p}^y should be divided by M because it represents the mean part expression?
  * In the alpha(...) function in Formula (7), s1 and s2 should be swapped.
* It would be nice to add more detailed comparison with most recent deep part-based models (e.g. [1], [2]) and show this work's novelty and strength.
* It's better to explain why the proposed method surpasses other methods considerably in some datasets (e.g. the Aircraft dataset) while only on part with other methods on some other datasets (e.g. the CUB dataset).

[1]: Few-Shot Classification with Feature Map Reconstruction Networks. CVPR 2021

[2]: Task Discrepancy Maximization for Fine-grained Few-Shot Classification. CVPR 2022


**Summary Of The Paper:**

This paper proposes an end-to-end part-based deep object parsing model for fine-grained few-shot classification. It learns a compact dictionary of salient part templates and performs multi-scale template matching during testing. Different from traditional part-based works, this paper uses feature map reconstruction as supervision to localize object parts and imposes a geometric prior on part locations during matching. This work provides good model interpretability and is competitive with very recent state-of-the-arts on several popular few-shot classification datasets.


**Summary Of The Review:**

I prefer to give a borderline accept score to the paper given that it clearly presents and provides a state-of-the-art part-based few-shot object classification method with good interpretability though some of its modules exist in previous works and some typos exist in the paper. It would be nice for the authors to make their code public to increase the potential impact of the work.

---

> ### Author Response · Authors · 2022-11-19
> **Initial response**
>
> Thank you very much for your constructive comments and questions.
>
> > Make notation easier to follow.
>
> Thank you for the suggestion. Going by this and reviewer cctR's question, we decided to rewrite the first 3 paragraphs of Sec. 3 to emphasize the notation that would be used throughout the section. We also modified Fig 2 and its caption to better correspond with this. All the remaining notation used have been rechecked and defined immediately before or after the equation they are used in.
>
> > Typos in equations.
>
> Thank you for pointing out the typos (which are accurate as you described). They have been fixed in the revision.
>
> > Comparison with existing part-based methods.
>
> In the supplementary of the submission, we presented a comparison of DOP with existing part-based methods, methods using reweighting or reconstruction criteria (Tab. 1) (In the revision, we added a pointer to the appendix under "Related Work"). TDM[2] uses reweighting of channels. While FRN[1] uses reconstruction criteria for query from the support features, it differs from DOP in that it does not share a dictionary of templates across all examples in the dataset which reconstruct single image features.
>
> > Better performance on Aircraft than on CUB.
>
> One possible reason for DOP's better performance on datasets like Cars and Aircraft is that these objects have less variance in part locations within the same class (since the objects are rigid). The use of part geometries allows for more accurate query-support comparison for our method. On the other hand, in images of birds and dogs, part geometries within the same class could have a higher difference (because of more variety of poses they could be photographed in), making comparison harder.
>
> > Implementation.
>
> In the supplementary materials of the revision, we have included a repository of our code with instructions to run experiments which we shall make public with the final version of the paper.

---

### Author Response · Authors · 2022-11-19
**Summary of changes**

We thank the reviewers very much for their constructive comments. We are encouraged they agree with the benefits of our method in improving fine-grained few shot recognition performance (all), the end-to-end trainability of our formulation (qRQa, SUHw) and our method's interpretability by locating salient object parts (SUHw, 47uq).

We added our initial responses including clarifications and changes made, under each reviewer’s main comment. Hoping for a productive discussion!

SUMMARY OF CHANGES\
-----------------------------------\
Based on the reviewers' feedback, we updated the paper and supplementary materials. Additionally, in the revised supplementary materials, please find attached a zip folder of the code to run our experiments. Instructions to run are included in the README.\
Besides minor grammatical changes and other minor changes to adjust space, following changes were made in the revision :

- Introduction : Updated "Contributions" paragraph (suggestion from 47uq) and made more explicit the relation between deep object parsing and feature noise reduction that can help in fine-grained few shot recognition task (under "Parsing"; question from cctR, SUHw).
- Related Work : Added pointer to supplementary appendix B where we systematically compare high level components of prior approaches in fine-grained few shot classification. Added FRN[a] to this comparison (suggestion from qRQa).
- Deep Object Parsing :
    - Rewrote first 3 paragraphs, and updated Fig. 2 for additional clarity (suggestions/questions from qRQa, cctR).
    - Fixed typographical errors mentioned by qRQa.
    - Notation fix in Eq 2 $[\phi\_c]\_{M(\mu)} \rightarrow \phi_{c, M(\mu)}$
    - Fixed expression for final estimate for $z_p$ (Eq. 6), with additional explanation and experiment added to supplementary (App. A, under "Differentiable Estimates").
- Experiments:
    - Added experiment to explain methodological components added on top of Prototypical Networks[b]. (This can answer the high level understanding related question from SUHw)
    - Moved ablation of instance-dependent reweighting and use of geometry for distance computation to the supplementary (App. D).

---

### Decision · Program_Chairs · 2023-01-20

**Decision:**

Reject

**Justification For Why Not Higher Score:**

Even positive reviewers (qRQa, 47uq) felt some relevant work was not discussed / compared with.  Reviewers also felt some parts of the paper was difficult to follow.  The authors did revise the paper to include more results and clarify some parts.  The AC feels that the changes are substantial and the paper warrants another review cycle.


**Justification For Why Not Lower Score:**

N/A

**Metareview: Summary, Strengths And Weaknesses:**

Summary: The paper proposes an end-to-end method for fine-grained few-shot classification by parsing an image into a collection of K parts from a shared dictionary.  The dictionary consists of K part template features and is learned from data to capture part instances from different categories at multiple scales.   Each parsed part in the image is composed of a 2D location and a part expression vector.  During parsing, multi-scale template matching is performed so as to optimize the feature map reconstruction error.  Experiments show that the proposed method is competitive with prior work on popular few-shot classification datasets while providing interpretability.

Strengths:
- The proposed method provides interpretability of results and has competitive performance
- The idea of using parts for few-shot classification is intuitive
- The proposed method can be trained end-to-end

Weaknesses:
- Some relevant work are not discussed and compared with (reviewer 47uq, cctR, qRQa)
- Some popular few-shot classification datasets were not evaluated on (reviewer 47uq)
- Reviewers found the paper to be difficult to follow at places